# Incidence of the El Niño–Southern Oscillation Cycle on the Existing Fundamental Niche and Establishment Risk of Some *Anastrepha* Species (Diptera-Tephritidae) of Horticultural Importance in the Neotropics and Panama

**DOI:** 10.3390/insects15050331

**Published:** 2024-05-04

**Authors:** Arturo Batista Degracia, Julián Ávila Jiménez, Anovel Barba Alvarado, Randy Atencio Valdespino, Mariano Altamiranda-Saavedra

**Affiliations:** 1Instituto de Innovación Agropecuaria de Panamá (IDIAP) PA 0739, C. Carlos Lara 157, Ciudad de Panama 0843-03081, Panama; 2Centro de Investigaciones Agroecológicas del Pacifico Central (CIAPCP-AIP), Panama Ciudad, Ciudad de Chitre 0601-00062, Panama; randy.atencio@gmail.com (R.A.V.); maltamiranda2@gmail.com (M.A.-S.); 3Facultad de Ciencias, Universidad Pedagógica y Tecnológica de Colombia, Avenida Central del Norte 39-115, Tunja 150003, Colombia; julianleonardo.avilajimenez@gmail.com; 4Institute of Agricultural Innovation of Panama/National Research System of Senacyt-Panama, Panama City 0816-02852, Panama; 5Grupo de Investigación Bioforense, Tecnológico de Antioquia Institución Universitaria, Medellín 050005, Colombia

**Keywords:** *Anastrepha grandis*, *A. serpentina*, *A. obliqua*, *A. striata*, distribution

## Abstract

**Simple Summary:**

Previous studies following this research allow us to contribute to the importance of potential distribution models of species and ecological niches. The *Anastrepha* genus, considered one of the most important at a quarantine level, demands a constant search for information to understand different ecological dynamics. We characterised the fundamental niches of four *Anastrepha* species in different El Niño–Southern Oscillation (ENSO) episodes (El Niño, El Neutro and La Niña) with ecological niche modelling in NicheA software 3.0. The results of a comparison of the ellipsoids that represent the fundamental niche existing for the species showed changes in the El Niño, El Neutro and La Niña episodes. Furthermore, when making a comparison between *Anastrepha* species and the different ENSO climatic episodes, we found that they share great environmental similarity between them. Finally, our results present different levels of risk of establishment of these species in the Neotropics, which will allow us to develop efficient plans for integrated pest management.

**Abstract:**

To compare the environmental space of four *Anastrepha* species in different ENSO episodes (El Niño, El Neutro and La Niña), we built ecological niche models with NicheA software. We analysed the fundamental niche and the combined establishment risk maps of these species developed with the ArcGisPro combine geoprocess. A comparison of the ellipsoids that represent the fundamental niche existing for the species showed changes in the El Niño, El Neutro and La Niña episodes. For *A. grandis* in the El Niño vs. El Neutro episodes, there was a Jaccard index of 0.3841, while the comparison between the La Niña vs. El Neutro episodes presented a Jaccard index of 0.6192. *A. serpentina* in the El Niño vs. El Neutro and La Niña vs. El Neutro episodes presented Jaccard indices of 0.3281 and 0.6328, respectively. For *A. obliqua*, the comparison between the El Niño vs. El Neutro and La Niña vs. El Neutro episodes presented Jaccard indices of 0.3518 and 0.7472, respectively. For *A. striata*, comparisons between the episodes of El Niño vs. El Neutro and La Niña vs. El Neutro presented Jaccard indices of 0.3325 and 0.6022, respectively. When studying the comparison between Anastrepha species and the different ENSO climatic episodes, we found that in the El Niño episode, the comparisons with the best environmental similarity were *A. obliqua* vs. *A. striata* and *A. obliqua* vs. *A. serpentina*, with higher Jaccard indices (0.6064 and 0.6316, respectively). In the El Neutro episode, the comparisons with the best environmental similarity were *A. serpentina* vs. *A. striata* and *A. obliqua* vs. *A. striata*, which presented higher Jaccard indices (0.4616 and 0.6411, respectively). In the La Niña episode, the comparisons that presented the best environmental similarity were *A. obliqua* vs. *A. serpentina* and *A. obliqua* vs. *A. striata*, with higher Jaccard indices (0.5982 and 0.6228, respectively). Likewise, our results present the risk maps for the establishment of these species throughout the Neotropics, allowing us to predict the level of risk in order to develop integrated pest management plans.

## 1. Introduction

Studies on how environmental conditions vary at different temporal and spatial scales have increased in recent years [1,2,3]. It is argued that the use of the word “niche” in an ecological context has theoretical bases, from 1917 to the present [4,5,6,7,8]. For this reason, niche studies are constantly updated and evolving, and insect species of the Tephritidae family, faced with this climatic variability, can vary their establishment patterns since they inhabit a wide variety of environments [9]. The family Tephritidae, to which the fruit fly belongs, is the most economically important, comprising approximately 4000 species distributed in tropical and subtropical areas [10,11,12]. Those known as fruit flies belong to various genera, including *Dacus*, *Rhagoletis*, *Ceratitis*, *Bactrocera*, *Anastrepha* and *Zeugodacus* [13,14]. Seven *Anastrepha* species are economically important in the Neotropical region due to their wide range of commercial host plants and distribution. These species include *Anastrepha ludens* (Loew) (Mexican fruit fly), *A. obliqua* (Macquart) (the West Indian fruit fly), *A. fraterculus* (Wiedemann) (South American fruit fly), *A. suspensa* (Loew) (Caribbean fruit fly), *A. serpentina* (Wiedemann) (Sapotaceas fly), *A. striata* Schiner (guava fly) and *A. grandis* (Macquart) (melon fly) [15]. These flies cause direct physical damage to the fruit pulp due to larvae and secondary damage caused by the entry of pathogenic microorganisms. There are also indirect implications, such as quarantine measures and export restrictions, thus limiting the development of various economies dedicated to fruit production [16,17,18,19].

Insects are vulnerable to extreme climate variability and fluctuations in climate components, such as temperature and precipitation, during certain periods of time [20]. Heat waves and seasonal temperature variations affect the development, movement, reproduction and behaviour of many organisms [21]. In recent years, insects have become a good model for evaluating the relationship between variations in environmental temperature and various traits of their life history [22]. This has allowed researchers to predict their responses to global warming and understand the physiological mechanisms that allow them to cope with temperature variations, such as changes in respiration, the use of antioxidants and certain proteins that protect them from heat [23]. Furthermore, the effects on their life cycles, genetic composition, hybridisation, distribution, and population abundance have been determined [24]. This is particularly true for members of the Tephritidae family, who inhabit a wide variety of environments [10,11,12]. In recent years, the production of fruits and vegetables has intensified considerably because of new patterns in the international economy (characterised by the globalisation of markets and technological development), changes in consumption patterns and competition between the different actors involved [25]. The harvested area of fruits and vegetables has grown worldwide in the last 30 years, reaching an average annual growth rate (AGR) of 3.33% between 1990 and 2019 [26,27,28].

Recent studies by the Atlantic Oceanographic and Meteorological Laboratory (AOLM) of the National Oceanic and Atmospheric Administration (NOAA) project variations in the El Niño–Southern Oscillation (ENSO) cycle due to the accumulation of greenhouse gases. The greenhouse effect predicts global climate impacts on temperatures and precipitation towards the second half of the 21st century [29]. Furthermore, a future pattern of changes in sea surface temperature (SST) equal to the La Niña phenomenon will increase the zonal temperature gradient and zonal advective feedback in the central equatorial Pacific, potentially increasing the frequency and amplitude of strong events of the El Niño phenomenon [30]. It is expected that, over the next few years, climate variability will cause alterations in the geographical distribution of insects because of the rearrangement of climatic zones [31,32]. Therefore, the magnitude of these impacts will be associated with the phenomenon of climate variability. The regions of Latin America, South America and the Caribbean are among the most vulnerable areas to climate variability since the majority of the species that live there are endemic or restricted to a specific tropical ecosystem [33,34,35,36]. Evaluations of how the environmental space available for a species can change in the face of climate variability can be used to improve integrated pest management (IPM) plans and production systems in different countries. Therefore, it is relevant to understand the impact of the ENSO on the ecological niche of *Anastrepha* species in the Neotropics and Panama. Due to the above, we have developed work on the potential distribution of these species in order to form groups of recordings on this topic and strengthen their applications in the scientific field [37]

## 2. Materials and Methods

### 2.1. Investigation Area

The study area was defined in the American Neotropics due to the economic importance of fruit activities and the occurrence of the species to be studied [8,38,39,40]. For spatial analyses, we used the ecoregion maps of the World Wildlife Fund (WWF), and the extension area between latitudes 30° N and 30° S was defined [37,41,42,43,44,45].

### 2.2. Species Presence

We used four species of the genus *Anastrepha* as study models (*A. grandis*, *A. serpentina*, *A. obliqua* and *A. striata*) due to their importance in horticulture at the Neotropical level and the need to understand their environmental requirements [37,46,47,48]. We obtained records of the presence of these species from the following sources: the Global Biodiversity Information Facility (GBIF), Species Link, the Centre for International Agricultural Bioscience (CABI) and the National Directorate of Plant Health of the Ministry of Agricultural Development (MIDA) [37,49,50].

### 2.3. Climate Data 

We analysed climate information in the Pacific Ocean from agencies specialising in atmospheric variability phenomena, which are the cause of uncertainty in environmental patterns today [37]. The agencies included NOAA in the United States (National Weather Service, Los Angeles, CA, USA, 2018), the Australian Government Meteorological Office (Bureau of Meteorology, Melbourne, Australia, 2018) and the Tokyo Climate Center in Japan (Japan Meteorological Agency, Tokyo, Japan, 2019). The information analysed from 2000 to 2019 allowed these 3 agencies to reach a consensus: 5 El Niño episodes, 3 Neutral episodes and 6 La Niña episodes [37]. These episodes were characterised by four rasters (minimum, maximum, medium and range), for which each of the environmental layers included the improved vegetation index (EVI, monthly Modis-Terra MOD 11C2v006), the temperature of land surface (LST, Modis-Terra MOD 11C3v006 monthly), the near-real-time precipitation rate (NRTPR, 3 h TRMM 3B42RTv7) and the normalised differential vegetation index (NDVI, Modis-Terra MOD 11C2v006 monthly). We selected these variables based on knowledge of the biology and natural history of the species, and 16 environmental layers were created for each episode with a spatial resolution of 0.25° or 25 km at the equator, allowing us to evaluate the behaviour of the ENSO cycle episodes [51,52,53,54,55,56,57,58,59].

### 2.4. Characterisation of the Existing Fundamental Niche

To compare the environmental space of the four *Anastrepha* species in the different ENSO episodes (El Niño, El Neutro and La Niña), we built ecological niche models with NicheA software ver 3.0, defining them as ellipsoids of minimum volume on an environmental background represented by a cloud of points in a three-dimensional grilled plane. NicheA first represents the studied space and constructs ellipses with the available occurrences as approximations of all the suitable environments of a species [60,61]. By projecting this ecological background with the 16 environmental variables of each episode, 8 datasets were constructed using the following criteria for selection: all variables (set 1); a Pearson’s correlation coefficient analysis to reduce collinearity (set 2) [57], where variables with a correlation value >|0.8| were removed using the corrplot package of R 3.6.0 statistical software [58,59]; a Jackknife analysis in MaxEnt [62] to assess the individual contribution of variables without spatial autocorrelation to the models, which included variables contributing ≥80% (set 3) [59]; variables with a variance inflation factor (VIF) <10 (set 4); all variables related to the NDVI (set 5); all variables related to the EVI (set 6); all LST-related variables (set 7); and all variables related to precipitation (set 8). The environmental dataset that met the selection of optimal parameterisations according to the Kuenm package [63] was sectioned to represent the environmental space of the species in each climatic episode. Subsequently, these were cut to the extent of the Neotropical region, being individually normalised to avoid the effect of the scale of each of them in the 3D representation (sequence in NicheA software [64]: >> Toolbox >> Utility Functions >> Normalisation/standardisation of variables), and with these, a principal component analysis (PCA) was created (sequence in NicheA software: NicheA >> Toolbox >> Background data > Component analysis main). Finally, this PCA was uploaded to NicheA to generate the background in the 3D representation (sequence in NicheA software: NicheA >> Toolbox >> Background data >> Draw Background Cloud Folder).

When characterising the niches of the species in each of the ENSO episodes, the occurrences were loaded (sequence in NicheA software: NicheA >> Toolbox >> Niche appearance >> Generate N(s) from occurrences) to design the corresponding folder that contained the niche attributes for each species in each context. Afterwards, the resulting folder was loaded in the background (sequence in NicheA software: NicheA >> Toolbox >> Niche Appearance >> Open N(s)) to graphically describe the realised niche through a minimum-volume ellipsoid representing the realised niche, and the points represent the environmental values of the episodes. To evaluate whether there were changes in the niches of the species, in the different episodes evaluated, the environmental spaces of each of these species built in the different episodes were set up on the background of the El Neutro episode, which was taken as a reference for all species comparisons. The result was three ellipsoids of minimum volume for each species: one for the El Niño episode, one for La Niña and another for average Neutral conditions. The comparison was performed using the niche overlap function (sequence in the NicheA software: NicheA >> Toolbox >> Niche Analysis Tools >> Quantify Niche Overlap), which performs paired contrasts by calculating the volumes of the ellipsoids and the portion of these that overlap in the multidimensional space. In addition, a modified Jaccard index was used to quantify niche overlap.

### 2.5. Establishment Risk Maps

Establishment risk maps were created with the product of binary maps that represented the potential distribution of each species in each climate episode [37,65,66,67,68]. By developing combined maps using the ArcGisPro combine geoprocess [69,70], this process generated a table of information for each created raster. Using the combinations of the Neutral episode, which is the one that presents the greatest number of combinations, the list of classes was produced and used to reclassify the three rasters using the reclassify geoprocess. An attempt was made to establish an order based on the names of the species (*grandis*, *obliqua*, *serpentina* and *striata*) according to the order in which they were combined. This established 16 possible combinations, which were then numbered for reclassification. New reclassification values were obtained, with which the shapes or layers were reclassified. When converting the classified raster layers to vector or shape file format to better work on the arrangement or grouping of classes, the raster to polygon geoprocess was used. When converting to vector data or shape files, thousands of polygons were generated for the 16 classes. After obtaining the shape files, the generated polygons were dissolved using the dissolve geoprocess. New fields were then added to the data tables of the three dissolved shapes to describe the 16 classes and to create new maps for the Neotropics and Panama [71,72,73].

## 3. Results

The comparison of the ellipsoids that represent the fundamental niche existing for the species showed changes in the El Niño, El Neutro and La Niña episodes. For *A. grandis* in the El Niño vs. El Neutro episodes, there was a Jaccard index of 0.3841, while the comparison between the La Niña vs. El Neutro episodes presented a Jaccard index of 0.6192 (Figure 1A). *Anastrepha serpentina* in the El Niño vs. El Neutro and La Niña vs. El Neutro episodes presented Jaccard indices of 0.3281 and 0.6328, respectively (Figure 1B). For *A. obliqua*, the comparison between the El Niño vs. El Neutro and La Niña vs. El Neutro episodes presented Jaccard indices of 0.3518 and 0.7472, respectively (Figure 1C). For *A. striata*, comparisons between the episodes of El Niño vs. El Neutro and La Niña vs. El Neutro presented Jaccard indices of 0.3325 and 0.6022, respectively (Figure 1D).

When making the comparison between *Anastrepha* species and the different ENSO climatic episodes, we found that in the El Niño episode, the comparisons with the best environmental similarity were *A. obliqua* vs. *A. striata* and *A. obliqua* vs. *A. serpentina*, with higher Jaccard indices (0.6064 and 0.6316, respectively) (Figure 2A). In the El Neutro episode, the comparisons with the best environmental similarity were *A. serpentina* vs. *A. striata* and *A. obliqua* vs. *A. striata*, which presented higher Jaccard indices (0.4616 and 0.6411, respectively) (Figure 2B). In the La Niña episode, the comparisons that presented the best environmental similarity were *A. obliqua* vs. *A. serpentina* and *A. obliqua* vs. *A. striata*, with higher Jaccard indices (0.5982 and 0.6228, respectively) (Figure 2C).

When analysing the risk maps for the neotropics, in the El Niño event, the high risk levels in the combination of *A. grandis*, *A. obliqua* and *A. striata* occurred in central Mexico, central Guatemala, the western coastal area of Colombia, eastern Brazil, coastal areas of Peru and the central area of Argentina. The combination of *A. grandis*, *A. serpentina* and *A. striata* occurred in western Guatemala, central Colombia, southern Peru, western central Bolivia, southeastern Brazil, southern central Uruguay, central Chile and the central part of eastern Argentina. The combination of *A. obliqua*, *A. serpetina* and *A. striata* was present in points in the north, west and south of Mexico, the west coast of Guatemala, north Nicaragua, north Costa Rica, central Panama, north Colombia and Venezuela, central and southern Brazil and northern central Argentina. The very high risk level of establishment in all combinations of *A. grandis*, *A. obliqua*, *A. serpentina* and *A. striata* was present in southeastern Mexico, Guatemala, Belize, northern Honduras, Nicaragua, northern Costa Rica, much of Panama, Colombia, Venezuela, Ecuador, Peru, Guyana, Suriname, Brazil, southern Bolivia, Paraguay, northern Argentina and Uruguay (Figure 3A).

In the El Neutro event, a high risk of establishment was presented by the combination of *A. grandis, A. obliqua* and *A. serpentina* in coastal areas of the Yucatán and western Mexico, central Guatemala, central eastern Panama, northern points in Colombia and Venezuela, northern and southern Brazil, much of Guyana, and southern Suriname. The combination of *A. grandis, A. obliqua* and *A. striata* occurred in central Mexico and Guatemala, northern Peru, eastern Brazil, southern Bolivia and a large part of central western Argentina. The combination of *A. grandis, A. serpentina* and *A. striata* occurred in northwest Guatemala, southern Peru, much of Uruguay, and eastern Argentina, and the combination of *A. obliqua, A. serpentina* and *A. striata* occurred in much of Mexico, northern Guatemala, Belize, Honduras, central Salvador and Nicaragua, central Costa Rica, western central Panama, northern and central Colombia and Venezuela, western Ecuador, the eastern centre of Brazil and eastern Argentina. A very high risk level of establishment in the Neotropics was found with the combination of the four species in southern Mexico, central Guatemala and Belize, a large part of Honduras and Nicaragua, central Costa Rica and Panama, a large part of central and south of Colombia and Venezuela, central Guyana, Suriname, French Guiana, north-central and southern Brazil, east-central Peru, north-central Bolivia, southeastern Paraguay, north-central Argentina and northwest Uruguay (Figure 3B).

In the La Niña event, a high risk of establishment was presented by the combination of *A. grandis*, *A. obliqua* and *A. striata* in south-central Mexico, central Guatemala, southwest Colombia, northeast and coastal areas of Peru and Chile, southwest Brazil, much of Argentina, and west-central Uruguay; by the combination of *A. grandis*, *A. serpentina* and *A. striata* in central Chile, east-central Argentina, southern Uruguay and Brazil; and by the combination of *A. obliqua*, *A. serpentina* and *A. striata* in coastal and central areas of Mexico, coastal areas of Guatemala, Salvador, northern Nicaragua, central and northwest Costa Rica, central Panama, central-northern Colombia and Venezuela, southwest Ecuador, central-west and east Brazil, southern Bolivia, much of Paraguay, and northern Argentina. Combinations of a very high risk of establishment with all species were found in central and southern areas of Mexico; a large part of Guatemala, Belize, Honduras, Nicaragua, Costa Rica and Panama; northern and southern Colombia and Venezuela; a large part of Guyana, Suriname, and French Guiana; much of north-central and southern Brazil; Peru; Bolivia; southern Paraguay and east-central Brazil (Figure 3C).

By analysing the cut for Panama, it was found that medium risk levels of establishment were presented in the El Niño episode with the combination of *A. grandis* and *A. serpentina* in western Panama (Bocas del Toro). A high irrigation level was observed with the combination of *A. grandis*, *A. obliqua* and *A. serpentina* in the coastal centre in Bocas del Toro, the north of Colon, and San Blas; with the combination of *A. grandis*, *A. serpentina* and *A. striata* in the northwest part of Chiriquí and southwest Bocas del Toro; and with the combination of *A. obliqua*, *A. serpentina* and *A. striata* in central-west Bocas del Toro, south-central Chiriquí, the central zone in Veraguas, Herrera, Los Santos, and coasts of Coclé and Panama Oeste. Very high establishment risk levels were found with all combinations of species in the provinces of Darién, Panamá, Colon, Panamá Oeste, north of Coclé, north and south of Veraguas, south of Los Santos, and much of Bocas del Toro and Chiriquí (Figure 4A).

In the El Neutro episode, the level of low risk of establishment was presented for *A. serpentina* in the southwest of Bocas del Toro, and a medium risk level was presented by the combination of *A. grandis* and *A. striata* in the north centre of Los Santos, south of Herrera and the central coast in Panama. A high level of establishment risk was presented by the combination of *A. grandis*, *A. obliqua* and *A. serpentina* in east-central Bocas del Toro, north and south-central Veraguas, southwest and northeast Colon, north-central Panama and San Blas, and some points of north and south Darién and by the combination of *A. obliqua*, *A. serpentina* and *A. striata* in southwest and east Bocas del Toro, north-central and south Chiriquí, central Veraguas, northwest Coclé, south Los Santos, the coastal centre in Panama, and northwest Darién. A very high risk of establishment in all species occurred in a large part of the territory of Chiriquí, Bocas del Toro, Veraguas, Coclé, Herrera, Los Santos, Western Panama, southeastern Panama and north-central Darién (Figure 4B).

In the La Niña episode, a low risk of establishment was presented by *A. serpentina* in the southwest of Bocas del Toro, and a level of medium risk was presented by the combination of *A. grandis* and *A. striata* in the centre of Coclé. A high irrigation level was presented by the combination of *A. grandis*, *A. obliqua* and *A. serpentina* in northeast and west Colon, central San Blas, points in central, east, and west Panama, and north-central Darién. A very high risk of establishment for all species occurred in a large part of Bocas del Toro, Chiriquí, north and south Veraguas, south Los Santos and Herrera, a large part of Coclé, Panama Oeste, Panama and Darién (Figure 4C).

## 4. Discussion

Changes in the distribution and niche of species in relation to climatic variability and anthropogenic effects are occurring in leaps and bounds; therefore, studies on this research topic are becoming increasingly important [74,75,76,77]. We characterised the changes in the existing fundamental niches for four species of the genus *Anastrepha* with occurrences during El Niño, El Neutro and La Niña episodes and found that these species have the potential to occupy different ecological spaces depending on the climatic ENSO scenario [37,78,79,80]. It is possible that aspects inherent to the biology of different *Anastrepha* species, such as the interaction with biotic variables, such as host plants, competing organisms, parasitoids, and predators, and local abiotic elements of the area accessible for the establishment of the insect, influence the niche breadth in response to the climatic variations of a locality [22,76]. There is an information gap associated with the host of the native species of *Anastrepha* in Central America since these native species have been little studied and their parasitoids are unknown [81]. There are a limited number of investigations that evaluate biotic and abiotic interactions; in Panama they do not exist, and in other regions of the Neotropics they are few (Mexico and Colombia) [82]. Our results for the four species showed changes in the configuration of the ellipsoids, as these species can adapt to survive. With these environmental variations regulating their physiological functions, depending on the ecological zone and according to the climatic episode, they could compete for resources in certain locations that they were not previously established [22]. The results demonstrated that the niches of the species evaluated share certain similarities in the shape and size of the occupied environmental space represented by the ellipsoids. These overlaps increase in El Neutro vs. La Niña events, in which these species manage to improve their fitness mainly via reproduction (longevity, better oviposition and fertile eggs) [77,83,84] and by being located in geographical areas in the Neotropics with stable temperature and environmental humidity conditions and with great biodiversity of specific and temporary host plants [85,86,87]. El Neutro vs. El Niño events presented similarities in the niche volume, which changed, decreasing the overlap. These ellipsoids represented only a portion of the fundamental niche due to the increase in temperatures and dry and arid conditions (such as desert areas, steppes or the Andes Mountain range), with scarce rainfall that restricts the expansion of species by affecting their reproductive biology at various stages (death of adults) and a decrease in the range of specific host plants [77,88,89]. On a practical level, these changes presented in the different climatic events will guide and strengthen decision making in IPM plans in the case of various species of fruit flies of the genus *Anastrepha*, which are of interest to various institutions associated with pest management at the international level [90]. Some fruit fly species, such as *A. striata*, can present a certain plasticity to changes in climatic conditions, maintaining or expanding their environmental space in different episodes and becoming a more competitive species for resources [22,91].

The overlap between the studied fruit fly species during the different ENSO episodes presented niches within an environmental space defined according to their ellipsoids. Some presented greater or lesser overlap, indicating that a certain number of species are affected during the El Niño and El Neutro episodes by competing for the same trophic resources if they maintain the same specific or alternative host plants and if they are influenced by the same climatic conditions (temperatures between 20 and 30 °C, average humidity of 75%) [22,91]. Under these abiotic and biotic precepts, *A. obliqua* and *A. striata* maintain their greater overlapping capacities in different geographical areas based on Hutchinson’s theory of duality, according to the ellipsoid corresponding to the species [89,92]. In the La Niña episode, species such as *A. serpentina* presented changes in the overlap, which could be due to high temperatures (35 °C), greater humidity and excessive precipitation at an ecological level [92,93]. All the information generated and compiled allows contingency plans to be organised for the control of insect pests, thereby optimising regional economic resources [94]. Furthermore, previous studies demonstrate that these species have a high degree of environmental suitability in the neotropics, thus corroborating the present study [37].

Databases of insect species at the international level need to establish projects that maintain updated research on this topic to strengthen the preventive control of these *Anastrepha* species [95,96,97,98]. The maps that present a very high risk of the establishment of all fruit fly species in the El Niño episode agree with the low levels of precipitation and average temperatures in tropical and subtropical zones. This could be due to the climatic variability that causes high levels of climatic, social and economic uncertainty in the American geography, and the availability of fruit hosts at the time that would increase the establishment risk [98,99,100]. As these climatic variables are among the most important at a biological level for *Anastrepha* species, they tend to influence their expansion or contraction on an ecological level, which will affect fruit plots and commercial export crops at harvest and postharvest, which are the dates of the greatest commercialisation [21,77]. The high degree of climate uncertainty in recent years with the El Niño and La Niña phenomena projects various in-depth analyses regarding the ENSO episodes that justify the study of models in short periods due to the constant changes in the temperatures of the Pacific Ocean [21,99]. Apparently, these changes in climatic patterns (precipitation and stable, average temperatures) increase the level of establishment risk for *Anastrepha* species, enabling them to find suitable areas for their reproduction [21,22,23,24,25,26,27,28,29,30,31,32,33,34,35,36,37,38,39,40,41,42,43,44,45,46,47,48,49,50,51,52,53,54,55,56,57,58,59,60,61,62,63,64,65,66,67,68,69,70,71,72,73,74,75,76,77,78,79,80,81,82,83,84,85,86,87,88,89,90,91,92,93,94,95,96,97,98,99,100,101,102]. The establishment risk maps presented will allow institutions, researchers and producers to be guided in specific areas of the Neotropics to design control plans to prevent the potential establishment of their populations according to a greater or lesser level of intensity of the environmental variable on site [100,102,103].

The fly species evaluated here have a high risk of establishing themselves throughout Panama due to a more stable climate throughout the different episodes of the El Niño phenomenon and its limited territorial geography with a fairly homogeneous flora that facilitates the expansion of these species, except in very dry and warm places in certain months of the year when the species *A. grandis* would not thrive [37,104,105]. These results indicate that the fruit fly species of the genus *Anastrepha* do not have geographical barriers in this region of the world, but their dispersion is limited by small changes in the availability of environmental space caused by the environmental variability of the ENSO cycle. This can be analysed with niche studies on small scales, which could help make better prevention decisions that alert the institutions responsible for phytosanitary safety and food safety policies of a country and thus optimise time and money [106].

## 5. Conclusions

The results of this research make it possible to guide and manage preventive phytosanitary plans in the IPM of species of fruit flies of the genus *Anastrepha* throughout the Neotropics in the face of climatic variability of the ENSO cycle. In addition, this research guides future basic and applied research work on other insect pests that attack fruit trees of interest to the international scientific community whose studies are limited to niche topics. In recent years, climate variability has changed the perspective in the field of research, and this methodological approach helps to materialise pest control plans throughout the world.

## Figures and Tables

**Figure 1 insects-15-00331-f001:**
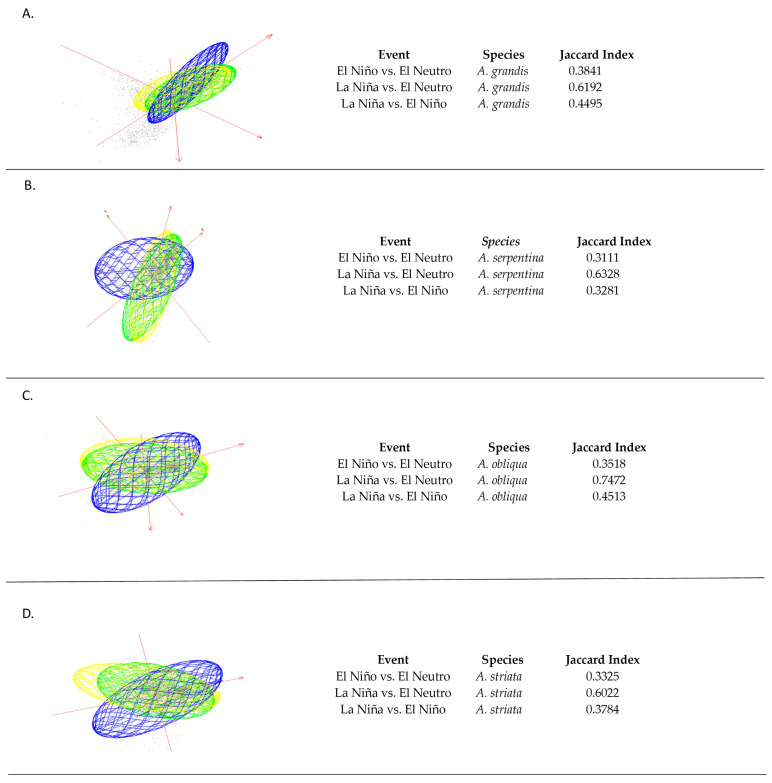
Estimated ecological niche models for *Anastrepha grandis* (**A**), *A. serpentina* (**B**), *A. obliqua* (**C**) and *A. striata* (**D**) and their superposition in environmental space. The axes are the main components of the 16 bioclimatic variables that describe the background of average conditions of the projection area, and the ellipsoids represent El Niño (blue), El Neutro (yellow), La Niña (green) and niches in the respective average conditions.

**Figure 2 insects-15-00331-f002:**
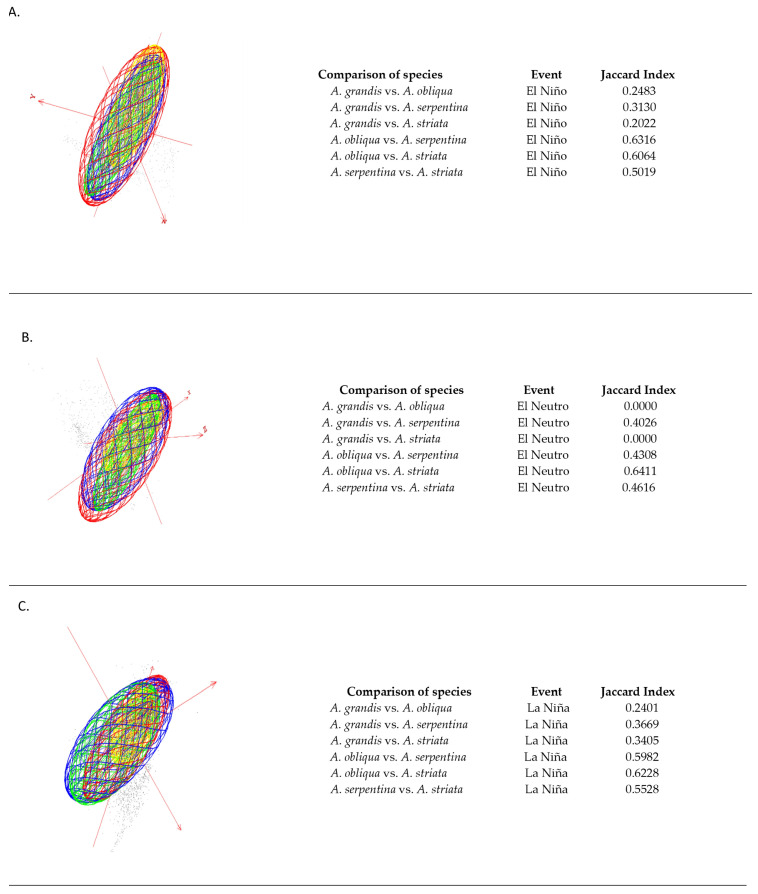
Estimated ecological niche models for *Anastrepha* species and overlap in environmental space. The axes are principal components of 16 bioclimatic variables that describe the background of average conditions of the El Niño (**A**), El Neutro (**B**) and La Niña (**C**) projection areas, and the ellipsoids represent the species *A. grandis* (yellow), *A. serpentina* (green), *A. obliqua* (blue) and *A. striata* (red) under the respective average conditions.

**Figure 3 insects-15-00331-f003:**
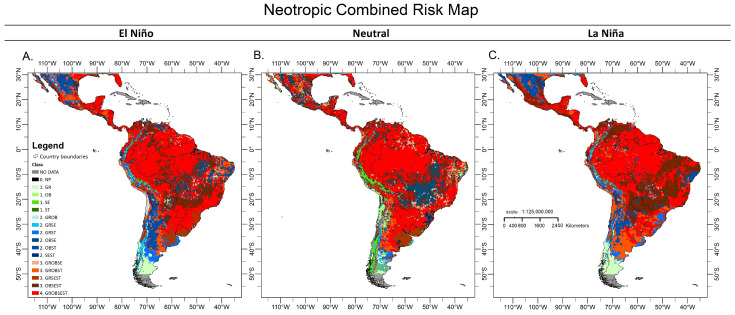
Combined risk maps of establishment for *Anastrepha* species in the American Neotropics, in episodes of El Niño (**A**), El Neutro (**B**) and La Niña (**C**). The legend specifies the following: drawing—border of countries, colours—*Anastrepha* species, numbers—level of risk of establishment, and united letters—different species. Synthesis: grey—No Data, black—0, NP—No Presence; letters: GR—*A. grandis*, OB—*A. obliqua*, SE—*A. serpentina*, ST—*A. striata*. Numbers with different colours represent the level of establishment risk of the species: 1, green—low establishment risk; 2, low to high shades of blue—medium establishment risk; 3, low to high shades of orange—high establishment risk; 4, red—very high establishment risk.

**Figure 4 insects-15-00331-f004:**
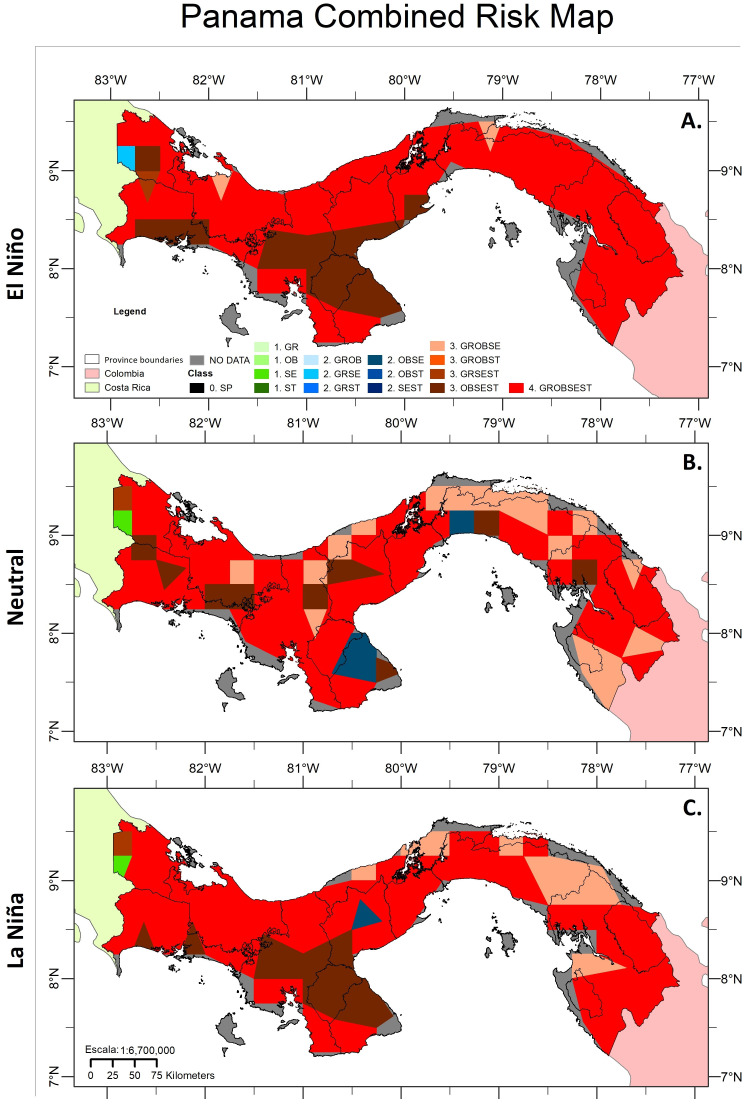
Combined maps of risk of establishment for *Anastrepha* species in Panama in episodes of El Niño (**A**), El Neutro (**B**) and La Niña (**C**). The legend specifies the following: white quadrant—border of provinces, pink quadrant—Colombia, cream quadrant—Costa Rica, colours—*Anastrepha* species, numbers—establishment risk level, joined letters—the different species. Synthesis: grey colour—No Data; black colour—0; SP—No Presence; letters: GR—*A. grandis*, OB—*A. obliqua*, SE—*A. serpentina*, ST—*A. striata*. Numbers with different colours represent the level of establishment risk for the species: 1, green—low establishment risk; 2, low to high shades of blue—medium establishment risk; 3, low to high shades of orange—high establishment risk; 4, red—very high establishment risk.

## Data Availability

All data are contained in the article.

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
