# Peer review of "Incidence of the El Niño–Southern Oscillation Cycle on the Existing Fundamental Niche and Establishment Risk of Some Anastrepha Species (Diptera-Tephritidae) of Horticultural Importance in the Neotropics and Panama"

_insects, 2024, doi:10.3390/insects15050331_

Round 1
Reviewer 1 Report
Comments and Suggestions for Authors
Please see attached file.

Comments on the Quality of English LanguagePlease improve presentation by using more direct and clear language and sentences.
Author Response
Response to reviewer 1

Reviewer 2 Report
Comments and Suggestions for Authors
The manuscript adresses a very relevant information on niche of four species of Anastrepha (Diptera: Tephritidae).
The title is very long. The introduction in the abstracts coccupies the most part of the abstracts. I suggested new keywords.
In the introduction, authors must highlight the advances in the new article in relation to the one already published (Degracia, A.B.; Jiménez, J.Á.; Alvarado, A.B.; Valdespino, R.A.; Altamiranda-Saavedra, M. Evaluation of the Effect of the ENSO Cycle on the Distribution Potential of the Genus Anastrepha of Horticultural Importance in the Neotropics and Panama. Insects 2023, 14, 714. https://doi.org/10.3390/insects14080714).
The methodology and results are well described.
The discussion should include more specific works, giving a clearer idea of the differences and similarities with other studies, even if they did not use the same method. Although the work did not aim to evaluate climate change scenarios, I suggest including this information to increase the number of readers. I suggest that the authors discuss using the works below.
https://doi.org/10.1017/S0007485321000985
https://doi.org/10.1007/s13744-019-00741-1
https://doi.org/10.1111/1744-7917.12018
https://doi.org/10.3389/fphys.2022.991923
https://doi.org/10.1016/j.cropro.2021.105836
Highlight progress in relation to the last article published by your team.
The conclusion must be more in-depth.. The names of some of the authors are incorrect in the references.
The are more susggestions in the attached file.

Author Response
response reviewer 2

Round 2
Reviewer 1 Report
Comments and Suggestions for Authors
Thank you for your careful revision. I believe the paper is improved and now acceptable.
Author Response
Thank you for your finals comments
Reviewer 2 Report
Comments and Suggestions for Authors
The authors made all corrections. Please, italicize Zeugodacus (line 99) and Anastrepha (line 416).
Author Response
thank you for your finals comments. Applied
